# Insights into Active Site Cysteine Residues in *Mycobacterium tuberculosis* Enzymes: Potential Targets for Anti-Tuberculosis Intervention

**DOI:** 10.3390/ijms26083845

**Published:** 2025-04-18

**Authors:** Abayomi S. Faponle, James W. Gauld, Sam P. de Visser

**Affiliations:** 1Department of Biochemistry, Faculty of Basic Medical Sciences, Sagamu Campus, Olabisi Onabanjo University, Ago-Iwoye 120107, Nigeria; 2Department of Chemistry and Biochemistry, University of Windsor, Windsor, ON N9B 3P4, Canada; gauld@uwindsor.ca; 3Department of Chemistry, Memorial University of Newfoundland, St. John’s, NL A1C 5S7, Canada; 4Department of Chemical Engineering, Manchester Institute of Biotechnology, The University of Manchester, 131 Princess Street, Manchester M1 7DN, UK

**Keywords:** metal coordination, posttranslational modifications, redox catalysis, anti-tuberculosis drug development, oxidative stress, nitrosative stress, anti-mycobacterial agents

## Abstract

Cysteine, a semi-essential amino acid, is found in the active site of a number of vital enzymes of the bacterium *Mycobacterium tuberculosis* (*Mtb*) and in particular those that relate to its survival, adaptability and pathogenicity. *Mtb* is the causative agent of tuberculosis, an infectious disease that affects millions of people globally. Common anti-tuberculosis targets are focused on immobilizing a vital cysteine amino acid residue in enzymes that plays critical roles in redox and non-redox catalysis, the modulation of the protein, enzyme activity, protein structure and folding, metal coordination, and posttranslational modifications of newly synthesized proteins. This review examines five *Mtb* enzymes that contain an active site cysteine residue and are considered as key targets for anti-tuberculosis drugs, namely alkyl hydroperoxide reductase (AhpC), dihydrolipoamide dehydrogenase (Lpd), aldehyde dehydrogenase (ALDH), methionine aminopeptidase (MetAP) and cytochromes P450. AhpC and Lpd protect *Mtb* against oxidative and nitrosative stress, whereas AhpC neutralizes peroxide/peroxynitrite substrates with two active site cysteine residues. *Mtb* ALDH detoxifies aldehydes, using a nucleophilic active site cysteine to form an oxyanion thiohemiacetal intermediate, whereas *Mt*MetAP’s active site cysteine is essential for substrate recognition. The P450s metabolize various endogenous and exogenous compounds. Targeting these critical active site cysteine residues could disrupt enzyme functions, presenting a promising avenue for developing anti-mycobacterial agents.

## 1. Introduction

### 1.1. Mycobacterium tuberculosis and the Importance of Enzyme Function

*Mycobacterium tuberculosis* (*Mtb*, *M. tuberculosis*) is the causative microorganism of human tuberculosis (TB) diseases [1,2]. *M. tuberculosis* has and continues to infect millions of people worldwide and is found in virtually all countries [2]. In fact, according to the World Health Organization, TB is second only to COVID-19 as the top infectious global disease, with an estimated 10.6 million people with TB infections in 2022. This is due in part to the fact that *M. tuberculosis* is adaptive and can thrive in various anatomical sites of the host [3]. However, it mostly affects the lungs, but in serious cases, it affects other body parts as well, including the lymph nodes, the eyes, the bones and/or bone joints, and the gut [4]. People with “active TB” show symptoms such as coughing, sometimes accompanied by sputum or blood, chest pains, fever and/or weight loss. These symptoms, if not treated immediately, can be fatal. Notably, most infected people have “latent TB” and show no symptoms but risk the disease progressing to “active TB”. In addition to the challenges faced by infection latency in TB, it can cause complications in those with other diseases such as HIV/AIDS, due in part to drug-resistant *Mtb* strains. As a result, there is a clear and increased need to develop new and more effective anti-tuberculosis therapeutic drugs. To support this effort, scientific studies have focused on understanding essential enzymes of *Mtb* and particularly those *Mtb* enzymes that are used to drive its metabolic activities, its virulence, its defense mechanisms and its persistence. Many of these vital *Mtb* enzymes contain an active site cysteine residue, and pharmaceutical efforts have focused on inactivating these cysteine residues. Reviewing and providing insights into the active site cysteine in enzymes of the various metabolic processes in *M. tuberculosis* opens a new perspective of the biocatalysis and drug discovery/development community working in this area of human infectious disease.

### 1.2. Metalloenzymes as Potential Therapeutic Targets

Metalloenzymes have diverse and ubiquitous roles in pathogenic organisms and, therefore, play central roles in the propagation of many diseases, including tuberculosis. As a result, such metalloenzymes are often present as potential targets for therapeutic interventions [5]. Furthermore, the active sites of a number of these critical metalloenzymes possess one or more cysteine residues. Thus, a cysteinate residue in a protein often binds metal ions, and, for instance, in the cytochromes P450, it links the heme to the protein scaffold [6,7,8,9]. Moreover, in the cytochromes P450 the cysteinate axial ligand causes a push effect toward the heme and makes its active species highly active for oxygen atom transfer reactions such as substrate hydroxylation (aliphatic as well as aromatic), substrate epoxidation and desaturation [10,11,12,13,14], although recently even defluorination reactions have been reported [15]. In addition, cysteinate is part of the catalytic triad in cysteine proteases that are involved in the hydrolysis of peptides and the biodegradation of proteins [16,17,18]. Finally, cysteine residues in proteins can form disulfide linkages that stabilize the structure and conformation of the protein [19,20,21]. Clearly, cysteine residues play important roles in enzyme structure and function, and inactivating these residues may affect their function. As such, cysteine residues in enzymes have become potential targets for drug therapies against *Mtb* due to the fact that they are often crucial for catalytic activity of the metalloenzymes [22]. Moreover, these groups may be susceptible to modification or reaction, and, therefore, are ideal targets for developing novel anti-tubercular therapies.

Metabolic processes, such as the shikimate pathway, which produces the aromatic amino acids phenylalanine, tyrosine and tryptophan, or aromatic intermediates, such as chorismite or phenolic compounds, are required for the adaptability, viability and survival of plants and apicomplexan protozoans [23,24,25]. The enzyme of the first committed reaction of the shikimate pathway is the metalloenzyme 3-deoxy-D-arabino-heptulosonate 7-phosphate synthase (DAHPS), which catalyzes the conversion of phosphoenolpyruvate (PEP) and erythrose-4-phosphate (E4P) into inorganic phosphate and 3-deoxy-D-arabino-heptulosonate 7-phosphate [26]. Furthermore, it possesses two key active site cysteine residues. Due to the critical role of the enzyme and biosynthesis function, it represents a potential key target for inhibition or inactivation, with the focus being on disrupting the entire metabolic pathway of the organism. Fortunately, this pathway is not present in humans and, therefore, is an attractive target for anti-parasitic and anti-tubercular agents [27]. A recent finding, through computational studies, suggested that the two cysteine residues present in the active site of DHAPS are important for structural stabilization of the catalytic scaffold, as well as regulating its catalytic activity [28].

In this review, an overview of the roles and mechanisms of several key enzymes essential to the viability and pathology of *M. tuberculosis* are presented. In particular, the focus is placed on those enzymes with active site cysteine residues and their potential as targets for drugs and other therapeutic strategies against tuberculosis. More specifically, the roles, mechanisms and potential therapeutic strategies of alkyl hydroperoxide reductase (AhpC), dihydrolipoamide dehydrogenase (Lpd), aldehyde dehydrogenase (ALDH), the lyase methionine aminopeptidase (MetAP), and the cytochromes P450 are reviewed.

## 2. Alkyl Hydroperoxide Reductase C (AhpC)

### 2.1. Function and Significance of AhpC in M. tuberculosis

Alkyl hydroperoxide reductase C (AhpC) is involved in the oxidative stress defense of *M. tuberculosis* against reactive nitrogen species, peroxides and peroxynitrite, and contributes to the survival of the microorganism within macrophages [29]. It catalyzes the reduction of organic hydroperoxides, which are generated by the host cells upon infection, to the corresponding alcohols [30]. AhpC is dependent on the interaction between an adaptor protein AhpD, and (reduced) NADH for its full peroxidase activity against peroxides and peroxynitrites [31]. AhpC peroxidase activity has been found to aid the bacterial virulence and mechanism of resistance to anti-tubercular drug, isoniazid [32,33]. AhpC gene regulation is uniquely marked with periods of silencing and activation, which relates to *Mtb*’s ability for antioxidant defense, especially during dormancy or transmission to new hosts [34]. AhpC belongs to the peroxiredoxin family of enzymes with evolutionarily conserved cysteine within the catalytic site, where it forms sulfenic acid intermediate during the reduction process [35].

### 2.2. Role of Active Site Cysteine in AhpC Activity and Potential as a Drug Target

This family of enzymes comprises the typical 2-Cys peroxiredoxins that use two cysteine residues for the catalytic oxidoreductive reactions [35]. They usually exist as functional dimers, but they undergo structural rearrangement to the oligomeric form in their catalytic intermediate states. Unlike other Gram-negative bacteria peroxidoxins that have structural folds of a dimeric pentamer in the reduced state, the mycobacterial AhpC would rather assume a ring-shaped hexamer of dimers [36]. In general, these enzymes’ reaction involves a peroxidative cysteine to react with the peroxide/peroxynitrite substrates to form cysteine sulfenic acid, which is subsequently attacked by a second resolving cysteine to form a disulfide crosslink (Figure 1). Each cysteine residue is located on a subunit of the dimer. The inter-subunit disulfide bond is reduced by associated proteins/oxidoreductases, again returning the peroxidative cysteine to the initial state.

In *Mtb*AhpC, the reducing equivalents for reducing the disulfide crosslink are supplied by AhpD, which is a thioredoxin-like protein present in a small number of organisms [31]. A third cysteine, whose catalytic role has not yet been determined, has been implicated in the catalytic mechanism of *Mtb* AhpC [37,38]. In contrast to other peroxiredoxins that feature local unfolding of the active site that exposes the peroxidative cysteine to the solvent environment [35], the *Mtb*AhpC has the capability of an entire conformational flexibility of the helix α_1_ of the loop-helix region that bears the peroxidative cysteine. This rigid-body movement of the entire helix brings the peroxidative cysteine in close proximity to resolving cysteine for disulfide crosslinking (Figure 1) [36]. The resolving cysteine provides an important function, which is to prevent further oxidation of the cysteine sulfenate by peroxide to form inactive sulfinic (-SO_2_H) or sulfonic acid (-SO_3_H). As such, a high concentration of resolving cysteine is maintained at the region complementary to the region bearing the peroxidative cysteine [35]. The chemical transformation is distinct from cysteine dioxygenase that uses dioxygen on an iron center to convert a free cysteine amino acid to cysteine sulfinic acid products [39,40,41,42,43].

Since the *Mtb*AhpC-AhpD peroxidase system is vital in protecting *M. tuberculosis* against oxidative stress triggered by peroxides/peroxynitrites [31] and is not present in humans, it has potential for anti-tubercular drug interventions. Studies have shown that *MtbAhpC* gene is expressed in compensation for the lack of *KatG*, a gene that encodes the catalase-peroxidase system that detoxifies hydrogen peroxide, in isoniazid-resistant *M. tuberculosis* in which the *KatG* gene was deleted [37,44]. There is a tendency to alter the role of the reducing partner protein, the AhpD, but this may not completely destroy the peroxide defense. For instance, a competitive inhibitor which competes for the substrate binding site of AhpD protein was unable to completely suppress the in vitro activity of AhpC/AhpD, and there was still a small amount of AhpD when isoniazid-resistant *M. tuberculosis* was studied in infected mouse lungs [45]. Still, the most effective strategy is to target the *Mtb*AhpC to cause total loss of the oxidative stress defense mechanism in the pathogenic organism. Guimarães et al. showed that there are implications for structure-based drug design that will mitigate or arrest the completion of the catalytic cycle of *Mtb* AhpC [36]. This was because of the unique manner in which the peroxidative cysteine moved close to the complementary second cysteine, which involved moving the entire helix α_1_, which generates a cavity that can potentially accommodate an inhibitor.

## 3. Dihydrolipoamide Dehydrogenase (Lpd)

### 3.1. Role of Lpd in M. tuberculosis Metabolism

Dihydrolipoamide dehydrogenase (Lpd) has its catalytic roles strategically placed for central energy metabolism and/or biosynthetic functions. It is one of the three components of a set of multienzymes that include the pyruvate dehydrogenase (PDH), the branched chain ketoacid dehydrogenase (BCKADH) and the α-ketoglutarate dehydrogenase (αKGD) complexes, which exist in both eukaryotic and prokaryotic organisms [46], although the αKGD enzyme is lacking in *Mtb* [47]. Lpd is the last enzyme (E3) of the PDH and BCKADH complexes and contains a flavoprotein disulfide reductase, which catalyzes the oxidation of dihydrolipoyl cofactor with the help of NAD^+^, as an acceptor of the reducing equivalent (Figure 2) [48,49].

In *Mtb*, Lpd’s function is involved in three different enzyme complexes (PDH, BCKADH, PNR/P) in a well-coordinated manner: the Lpd, Dlat (dihydrolipoyl acyltransferase, E2) and AceE (E1) multi-components synthesize acetyl-coA in the PDH complex; the Lpd, Dlat and pdhABC complexes produce the branched-chain fatty acyl-coA in the BCKADH complex; and the Lpd, Dlat and AhpC/AhpD system detoxifies peroxides/peroxynitrites (PNR/P) [50].

The enzymatic products of these complexes are essential for the metabolic homeostasis, cellular survival and adaptability of *Mtb* in the macrophages, ensuring that the energy requirements of the organism are met continuously even when the host cells are in a nutritional deficient state. They also ensure that the organism continues to mount defense responses against host-induced oxidative and nitrosative stress, as well as the reduction in the over-accumulation of metabolites such as pyruvate, branched-chain amino acids and branched-chain keto acids, which can become potentially harmful to the organism itself.

### 3.2. Impact of Active Site Cysteine on Lpd Function and Drug Development

Dihydrolipoamide dehydrogenase (Lpd) operates a disulfide-based redox active center [51]. The disulfide crosslink is contributed by two active site cysteine residues that are evolutionarily conserved in many organisms [46,51,52,53]. The structure of Lpd’s active site is arranged in such a way that the electron flow from the dihydrolipoyl substrate to NAD^+^, via FAD (flavin adenine dinucleotide) is functionally related to the formation of NADH, which regenerates the disulfide crosslink to complete the catalytic cycle (Figure 3).

Lpd is a homodimeric enzyme, but each monomer contains four distinct domains which are sites for FAD binding and NAD^+^ binding. In addition, the structure contains a central domain and an interface domain [54]. The two active site cysteine residues are located in the FAD binding domain, where they interact with the tightly but non-covalently bound FAD. The interface domain has a conserved His-Glu motif, which donates a proton (H^+^) for the catalysis [55]. The NAD^+^ molecules are located on their own separate domain, which indicates the requirement for the enzyme to devise an electron flow in this manner. Disruption of this electron flow will negatively affect enzyme functions and thus the upstream metabolic pathways, leading to energy starvation and the accumulation of toxic metabolic intermediates such as pyruvate or lactate [56].

One of the two active site cysteine residues is substrate-binding (proximal) and the other interacts with the FAD molecule. Interestingly, it has been shown that the substrate-binding cysteinyl residue is more reactive than the distal one, which indicates that their thiol groups are chemically inequivalent [57]. As such, the proximal cysteine is more amenable to further reactions such as nitrosylation and sulfenation [58]. These chemical modifications could cause protein misfolding [59] and, in the case of Lpd, might result in loss of its enzymatic activity [60].

In *Mtb*, the electron flow arrangement is such that the first enzyme component that accepts electrons from NADH is Lpd (direction of flow of electrons: NADH → Lpd → DlaT → AhpD → AhpC → ROOH), which has its redox center regenerated in the process. As such, Lpd’s function is critical in this arrangement. Electron flow would be disrupted when there is impairment of function or even absence of any of the components [31]. Any chemotherapy agent that can target any of these components can disrupt the electron flow. Therefore, such an agent will have potential to attenuate *Mtb*’s survival because the organism becomes susceptible to oxidative and nitrosative stress induced by the host immune response [31,50]. It has been shown that chemical inhibitors like triazaspirodimethoxybenzoyls have the potential to selectively inhibit *Mtb*Lpd, with >100-fold selectivity compared to human Lpd [61]. Although both share some evolutionary conservation, there are variations in the active sites of the *Mtb*Lpd and human *Mtb* [49]. The differences in the active sites could be explored to design and develop useful species-selective inhibitors of Lpd, particularly those targeting the active site cysteine residues, as potent anti-tubercular therapeutic agents.

## 4. Aldehyde Dehydrogenase (ALDH)

### 4.1. Function of ALDH in Detoxification Pathways

Aldehyde dehydrogenases (ALDH) are a superfamily of NAD(P)^+^-dependent proteins that catalyze the oxidation of a wide range of aldehydes (aliphatic and aromatic, both endogenous and exogenous) to their corresponding more soluble carboxylic acids [62,63]. These enzymes are present in most organisms including prokaryotes, eukaryotes and *Archaea*. ALDH have various (non)catalytic functions that involve detoxification, biosynthesis, antioxidant functions, and structural and regulatory mechanisms, including roles in embryogenesis, development and neurotransmission, oxidative stress, and cancer [64,65]. There is more than one ALDH gene in humans [66,67]. Thus, the human ALDH-2 protein has many functions, including nitrate reduction [68,69] and the removal of toxic aldehydes. Accumulation of aldehydes can lead to protein/enzyme dysfunction, oxidative damage, the generation of reactive oxygen species, and lipid peroxidation, which causes the formation of more aldehydes [65,70]. *Mtb* is also known to have ten putative ALDH proteins. Bioinformatics analyses have associated them with seven different *Mtb* ALDH classes. Notably, only one of the *Mtb* ALDH proteins, *Mtb* Rv0223c, has been experimentally solved and was comparable to the human ALDH-2 because of their close sequence and structural similarity to each other [66].

In this case, they share three domains (catalytic domain, coenzyme domain and oligomerization domain) that align very highly when superimposed (Figure 2). However, *Mtb* Rv0223c is a monomer, in contrast to the octameric human ALDH-2 [71,72,73]. Despite this difference, both proteins retain the evolutionarily conserved cysteine and glutamic acid at identical positions within their active sites. The active site cysteine is directly involved in the detoxification mechanism of *Mtb* ALDH.

### 4.2. Involvement of Active Site Cysteine in ALDH Activity and Drug Targeting Strategies

Aldehydes make *Mtb* sensitive to nitric oxide (NO) and Cu, which are natural antimicrobial agents produced by the host immune cells [74,75]. Recently, evidence has been put together in a review that shows aldehydes of host origin have the ability to control intracellular growth of *M. tuberculosis* [76]. Limón et al. showed that the wild-type *M. tuberculosis* became susceptible to Cu toxicity when an exogenous aldehyde, *para*-hydroxybenzaldehyde (pHBA), was added to the mycobacterial strain cultures [77]. As such, the presence of some aldehydes may be important for controlling *M. tuberculosis* infections. Since the human and *Mtb* ALDH2 are highly homologous, it is presumed that both have similar catalysis. Importantly, the active site cysteine acts as a nucleophile that attacks the aldehyde functional group (the carbonyl carbon), resulting in the formation of an oxyanion thiohemiacetal intermediate, which is stabilized by amide/peptide nitrogen atoms of nearby amino acid residues [78,79]. This step is crucial in the aldehyde detoxification mechanism mediated by *Mtb*. Overall, the implications of a defective or dysfunctional *Mtb* ALDH will be apparent absence or perturbed catalysis leading to the buildup of aldehydes, which attenuate the growth of the bacteria within the macrophages.

There is potential to design and develop drugs that will interfere with this critical step in the catalysis of *Mtb* ALDH, particularly by preventing the active site cysteine from performing the nucleophilic attack (step 2 in Figure 4) upon cofactor binding. For instance, it has been shown before that disulfiram, a sobriety drug used to treat alcoholism, which was repurposed against *Mycobacterium tuberculosis* showed significant anti-tubercular activities against both multidrug- and extensively drug-resistant strains of the microbe in mice [80]. Also, there was a synergistic action of Cu ions and disulfiram in killing *Mtb*, which suggests that the antibacterial effects of the drug are copper-dependent [81]. While the mechanism of the bactericidal activity of the drug was unclear, another study had revealed the possibility of the drug inducing interactions between two active site cysteine residues, thereby forming a stable intramolecular disulfide bond [82] that halts catalysis. This is very insightful and opens a broad window of opportunities to develop and screen a wide range of inhibitors that have promising potential to interfere with the active site cysteine in *Mtb*. It is useful to research species-specific inhibitors that target the pathogenic enzyme, as well as co-agents that induce the innate host immune defenses, resulting in the accumulation of reactive aldehydes and thus leaving the enzyme susceptible to their actions. A novel chemotherapy strategy will be to induce physiological events that increase the amount of metabolic aldehydes that kill the bacteria while sparing the human host cells. By these means, the chemotherapy agents can be developed to perturb the catalytic role of the active cysteine of *M. tuberculosis*, leaving the accumulated aldehydes cytotoxic only to the invading pathogen.

## 5. Methionine Aminopeptidase (MetAP)

### 5.1. Role of MetAP in Protein Processing

Methionine aminopeptidases (MetAP) are present in all living organisms, including bacteria and humans, where they are involved in the translational modification of proteins in the cells. They are dinuclear metalloenzymes that rely on first-row transition metals as cofactors for catalytic activity. Specifically, these metalloproteases cleave the *N*-terminal methionyl residue from all newly synthesized proteins (Figure 5) [83,84].

This translational processing occurs in most nascent proteins and is an important process in endothelial cell growth and the differentiation for angiogenesis, which is the formation of new blood vessels, and cell cycle/proliferation. In humans, annihilation or suppression of this reaction would prevent the *N*-terminal residues from undergoing further modifications, leading to blockage of angiogenesis [5,85,86] and control of cell proliferation [87,88,89]. This signifies the importance of MetAP inhibitors as potential anticancer agents. Nonetheless, the focus here is to illuminate the structure and functions of mycobacterial MetAP in an effort to reveal the potential of developing species-specific inhibitors that target their active site cysteines. There are two classes of MetAP—namely type 1 (MetAP1) and type 2 (MetAP2)—that are present in all organisms. The difference between the two types is found in type 2, in which there is an approximately 60-residue insert within the α-helix fold of the catalytic domain [5,90,91]. While both types are present in eukaroytes and humans, the prokaryotes possess either one of the two types with several homologues [84]. There are subclasses, designated *a*, *b*, *c* and *d*, for type 1 MetAPs, with unique features in the *N*-terminal domains, such as the presence of a zinc finger domain in subclass *b* [92]. Two isoforms of the enzyme are found in *Mtb*, *Mt*MetAP1a and *Mt*MetAP1c; the latter possesses a highly conserved proline-rich *N*-terminal extension [85]. Both isoforms are enzymatically active and not functionally redundant [93]. There was mycobacterial growth when either one of them was over-expressed in the presence of inhibitors. However, knockdown of only *Mt*MetAP1a resulted in reduced growth, which suggests its essentiality in the viability of *Mtb* [93]. Therefore, efforts to develop anti-mycobacterial agents could be directed at the mitigation of the enzymatic activity of *Mt*MetAP1a.

### 5.2. Significance of Active Site Cysteine in MetAP Function and Drug Targeting Strategies

The active sites of MetAPs contain a dinuclear metal center, and the residues coordinating these metals are highly conserved in all organisms [5,90]. The substrate-binding pocket is surrounded by non-conserved but homologous amino acid residues. Figure 3A depicts the dinuclear metal coordination motif and the substrate-pocket residues of *Mt*MetAP1c. Importantly, a cysteine is found to be conserved among these residues in all MetAP1, although another residue, glycine, is present in the homologous position in MetAP2 [94,95]. Reddi et al. performed a sequence alignment test of *Mt*MetAP1c with other type 1 MetAPs from other species and identified the conserved Cys105 residue at the same location in all strains (Figure 3B) [96]. On the other hand, only the *E. coli* MetAP1 active site Cys70 has been revealed to participate directly in catalysis [97], while the Cys105 residue is required to play an essential role in substrate positioning. The active site Cys105 role may be harnessed in designing specific MetAP1 inhibitors. Reduced enzymatic activities of *E. coli* and human MetAP1s when the active site cysteine was mutated [94] may reinforce the plausibility of designing specific inhibitors for therapeutic interventions against pathogenic diseases.

The active site Cys105 can be covalently modified only by some specific inhibitors. The mechanism of selective modification is dependent on the stereochemistry of the cysteine and the presence of divalent metal ions [96]. Generally, MetAPs are capable of using any first-row transition metals for catalytic activity [98], although cobalt ions have been used frequently to reconstitute the enzyme in many studies [5]. Both the cysteine-specific inhibitory agent and cysteine must be in an orientation that favors a nucleophilic addition. The metal ions help to stabilize the complex through a bridging water molecule. Since MetAPs have the capacity to utilize a range of metal ions for catalysis, it may present some difficulties in designing potent inhibitors against them. However, a promising approach is developing species- and site-specific inhibitors that target the active site cysteine through covalent modification, disrupting its functions within the enzyme.

## 6. Cytochromes P450

The cytochrome P450s (P450s or CYPs) are enzymes present in most organisms, ranging from prokaryotes to eukaryotes, in which they mediate a wide range of metabolic reactions, including the biosynthesis and biodegradation of compounds in both endogenous and exogenous molecules [6,7,8,9,10,11,12,13,14,15,99,100,101,102,103,104]. The P450s are monooxygenases that insert a single oxygen atom derived from a dioxygen molecule into their substrates, thus changing the physicochemical properties of the substrates, as happens in the detoxification of xenobiotics, for instance. Generally, the P450 active site consists of a heme iron (porphyrin) cofactor, bound through a heme–thiolate interaction from an active site cysteinate residue, while the substrate binding pocket has important amino acid residues that relate to substrate positioning and proton and electron relay [105]. As an example, we show in Figure 4 the active site of a typical P450 enzyme, namely an extract from the protein databank file 6UPI [106,107] that represents the dicyclotyrosine-bound isozyme CYP121. The heme iron (grey) lies on the equatorial plane of the active site pocket and has a distal site, where the dioxygen and substrate bind, and a proximal site that links it to the axial Cys345 residue of the protein.

Usually, the P450 reaction cycle goes through a series of pre-catalytic events, which produce the active oxidant (known as Compound I, Cpd I) of the enzyme reaction, before returning to the enzyme resting state to complete the catalytic cycle [6,7,8,13,14,101,103,104,105,108]. Briefly, when the substrate binds into the P450 active site, a water molecule that occupies the distal site is displaced and leads to a change of spin state from low-spin to high-spin. Thereafter, the heme iron(III) is reduced to iron(II), where the electron is supplied by a reducing partner, e.g., NAD(P)H or P450 reductase. Next, the heme iron(II) readily binds dioxygen to form an adduct. A second electron is supplied by the redox partner that changes the iron-dioxygen into an iron-peroxo group. This heme peroxo intermediate is protonated to form the heme iron(III)-hydroperoxo complex and ultimately forms Cpd I through a subsequent protonation and heterolytic O–O bond cleavage that releases a water molecule. CpdI is a highly reactive intermediate with substrates and typically reacts through oxygen atom transfer. The Cpd I was an elusive species until Rittle and Green were able to characterize it spectroscopically in 2010 [109]. Several studies have shown the importance of the thiolate contribution to the reactive nature of Cpd I—the oxidizing equivalents are shared between the porphyrin architecture and the thiolate ligands [10,11,12,13,14,15,110,111,112,113]. The influence of the heme-thiolate is termed the “push effect” whereby the axial ligand thiolate promotes an increased electron density on the heme iron and the oxo atom, thereby enhancing the reactivity of Cpd I active oxidant of P450 enzyme. With enhanced reactivity occasioned by the push effect of the heme-thiolate, Compound I typically reacts with substrates through oxygen atom transfer reactions, including aliphatic and aromatic hydroxylation, sulfoxidation, epoxidation, dealkylation, decarboxylation and desaturation, among others [6,7,8,9,101,102,103,104,114,115,116,117].

Mycobacteria contain several CYP enzymes; *Mycobacterium tuberculosis* encodes 20 CYPs, which contribute to its survival and pathogenicity. For example, *Mtb* CYP121 is involved in the biosynthesis of mycocyclosin, an important natural compound in the bacterium that regulates cellular homeostasis and controls its survival and growth [118,119,120]. Additionally, the heme-thiolate has been shown to act as a redox switch that activates and deactivates P450 catalytic activity through the (reversible) formation of sulfenic acid in the presence of hydrogen peroxide and thus inhibits the enzyme function [10]. Also, the P450 2B4 isozyme nearly lost its monooxygenase activity when its active site cysteine was substituted with serine, emphasizing the vital role the heme-thiolate plays in the catalytic life of the P450s [111]. Again, the heme-thiolate plays a role in spin state and redox switching within the P450 catalytic cycle by ensuring low energy transition between the high- and low-spin states [113]. Largely, since the P450 active cysteine is *sine qua non* to P450 catalysis, it is an ideal target for enzyme function disruption, particularly in mycobacterial agents such as *M. tuberculosis*.

## 7. Conclusions

Cysteine, a semi-essential amino acid residue found in proteins, may be evolutionarily conserved in certain protein sequence regions due to needs such as the catalysis, regulation and modulation of protein and enzyme activity, protein structure and folding, its structural motif—particularly in cofactor metal coordination—and its role in posttranslational modifications of nascent proteins. The cysteinate group often participates in redox and non-redox enzymatic reactions and serves as a nucleophile. Cysteine is found in the active site of a wide range of enzymes, including those present in pathogenic organisms, in which they aid in propagating diseases such as tuberculosis, which has *M. tuberculosis* as the etiological agent. *M. tuberculosis* possesses enzymes that play roles in cellular defenses against oxidative and nitrosative stress caused by host-derived reactive oxygen/nitrogen species, e.g., peroxides and peroxynitrite, as well as removal of cytotoxic agents such as over-accumulated host cell aldehydes. These enzymes ensure *Mtb*’s survival, virulence and persistence in the host’s macrophages. The enzymes discussed in this review are involved in *Mtb*’s survival and adaptability. For example, alkyl hydroperoxide reductase (AhpC) and dihydrolipoamide dehydrogenase (Lpd) are parts of the enzymatic anti-oxidative arsenal that ensure *Mtb*’s viability; aldehyde dehydrogenase (ALDH) clears a wide range of aldehydes from both endogenous and exogenous sources which are toxic to its own cell, ensuring its survival. Methionine aminopeptidase (MetAP), a metalloenzyme that is active with many of the first-row transition metals, cleaves the *N*-terminal methionine during the translational modification that prepares the nascent protein for further protein modification, which is essential in the regulation of the cell cycle and cell proliferation, thus ensuring *Mtb*’s survival and virulence in the host cells. The cytochromes P450 mediate several important reactions involving both endogenous and exogenous compounds in the microorganism, ensuring its adaptability, survival and pathogenicity.

### 7.1. Implications for Drug Discovery

The active site cysteine of these *Mtb* enzymes is important for their enzymatic functions. The two active site cysteine residues in AhpC work in a coordinated manner to neutralize peroxide/peroxynitrite substrates. While the peroxidative cysteine forms cysteine sulfenic acid with the substrates, the resolving cysteine returns the former to the initial disulfide state. Interference or blocking of these reactions would halt the enzyme function. Any chemical agents that can target *Mtb* and alter this step in the catalysis are potential anti-tuberculosis interventions. Also, Lpd has two active site cysteine residues, which form a disulfide link and are involved in electron transport to NAD^+^ via FAD. Disruption of the electron flow would block the multienzyme complex function. This could be achieved by targeting the reactive substrate-binding cysteine of Lpd in the *Mtb* enzyme. Since this reaction is vital for the organism’s central intermediary metabolism, lack of it will lead to microbial death due to energy starvation and over-accumulation of toxic metabolic intermediates. Again, an inhibitor of this enzyme function is a promising anti-tubercular agent. ALDH detoxifies aldehydes with its active site cysteine. In ALDH catalysis, cysteine provides the thiol nucleophile that attacks the aldehyde carbonyl carbon to form an oxyanion thiohemiacetal intermediate in the oxidation reaction, leading to the formation of carboxylic acid. Interfering with the cysteine nucleophilic attack would block this crucial oxidative reaction. This is also a target for potential anti-tubercular agent. In *Mt*MetAP, the active site cysteine plays an essential role in substrate recognition and positioning, rather than being involved in direct catalysis, as found in *E.coli*MetAP. Indeed, this reveals the possibility of designing a site-specific compound that can block the enzyme’s ability to recognize its native substrates. Lastly, by targeting or blocking the catalytic contribution of the heme-thiolate moiety of the P450 enzymes, the metabolic processes mediated by them can be disrupted, thus providing a lasting solution to the problem of *Mtb* infections.

### 7.2. Challenges and Future Research Directions

The *Mtb* enzymes discussed in this review are essential for *Mtb*’s survival and pathogenicity. However, some of them are present and/or have overlapping functions in human host cells. The challenges lie in the efforts of being able to target microbial enzymes while sparing the human enzyme functions. Some of the microbial enzymes have evolved somewhat dissimilar mechanisms, perhaps to evade host cells’ defenses. When such differences are recognized, they may open the possibility of developing specific inhibitory agents. For instance, although AhpC are found in *M. tuberculosis* and humans, the *Mtb*AhpC-AhpD peroxidase system that is vital in protecting *Mtb* against oxidative stress is not present in humans, which makes it a potential specific target for drug discovery and development. Thus, it has become imperative to continually find not only target-specific inhibitors of enzyme functions but also those that are species-specific, leaving the human host enzymes intact.

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
