# Peer review of "Insights into Active Site Cysteine Residues in Mycobacterium tuberculosis Enzymes: Potential Targets for Anti-Tuberculosis Intervention"

_ijms, 2025, doi:10.3390/ijms26083845_

Round 1

Reviewer 1 Report

Comments and Suggestions for Authors

This review provides a concise and informative overview of the critical function of cysteine residues in key Mycobacterium tuberculosis (Mtb) enzymes. The focus on four specific enzymes (AhpC, Lpd, ALDR, and MetAP) and their dependence on active site cysteines for activity is well rationalized and provides a clear framework for understanding the potential for targeting these residues for drug development. However, the author should add CYP enzymes as they are also an essential part of regulation through cysteine residues, and author referenced many of their own articles in the reference section explaining the importance of cyp enzymes.
The article unmistakably highlights the importance of cysteine residues in Mtb viability, flexibility, and pathogenicity.  The descriptions of the roles of the selected enzymes and the mechanisms by which cysteine residues are involved are clear and easy to follow. The last remark regarding the possibility of designing anti-mycobacterial molecules by targeting these cysteine residues is an appropriate and solid conclusion derived from the above discussion.
The writing is clear, concise, and well-organized.  The article should be accepted after adding the CYP enzymes. 

Comments on the Quality of English Language

None

Author Response

Rebuttal letter to referee 1:

This referee is supportive of publication of our work in IJMS and writes: “This review provides a concise and informative overview of the critical function of cysteine residues in key Mycobacterium tuberculosis (Mtb) enzymes. The focus on four specific enzymes (AhpC, Lpd, ALDR, and MetAP) and their dependence on active site cysteines for activity is well rationalized and provides a clear framework for understanding the potential for targeting these residues for drug development. …. The writing is clear, concise, and well-organized.  The article should be accepted after adding the CYP enzymes.”

Reply: Thanks for reading our manuscript and the valuable comments made.

In addition, the following comment is made: “However, the author should add CYP enzymes as they are also an essential part of regulation through cysteine residues, and author referenced many of their own articles in the reference section explaining the importance of cyp enzymes. The article unmistakably highlights the importance of cysteine residues in Mtb viability, flexibility, and pathogenicity.  The descriptions of the roles of the selected enzymes and the mechanisms by which cysteine residues are involved are clear and easy to follow. The last remark regarding the possibility of designing anti-mycobacterial molecules by targeting these cysteine residues is an appropriate and solid conclusion derived from the above discussion.”

Reply: Thanks for the suggestion, we have now added a section on the P450 enzymes on how they may be relevant to Mtb drugs and therapies.

Reviewer 2 Report

Comments and Suggestions for Authors

The manuscript with the title “Insights into active site cysteine residues in Mycobacterium tuberculosis enzymes: Potential targets for anti-tuberculosis intervention” is a study carried out on vital enzymes of the bacterium Mycobacterium tuberculosis (Mtb) and how the mechanism of action of these enzymes may be influenced by the action of drugs in the treatment of tuberculosis.
The importance of the study is very high considering the large number of infected people. 
The Cys residue from the catalytic site of four enzymes, its role as a nucleophile and the mechanism of action of those enzymes has been studied. It was also noticed how these can be targeted in therapy by specific drugs. The four Cys-enzymes focused are Alkyl hydroperoxide reductase, Dihydrolipoamide dehydrogenase, Aldehyde dehydrogenase and Methionine aminopeptidase. These enzymes are responsible for the viability and adaptability of Mtb and the infection of host organisms.
The involvement of the Cys residue in substrate recognition and bonding, in the mechanism of enzymatic catalysis is explained, clearly resulting in how it can be interfered with anti-tuberculosis drugs. Interference in the enzymatic catalysis mechanism with different chemical agents with an inhibitory role of the enzyme could be the way to finding new anti-tuberculosis drugs.
I think the study is well done. I have some recommendations.
In chapter 5.2. Significance of active site cysteine in MetAP function and drug targeting strategies, row 347 is discussed in Figure 4(A). This figure cannot be found in the manuscript. 
The bibliography is suitable for the topic discussed, but old enough for a review. Unfortunately, only 12 of the total cited references are from the last 5 years. I strongly recommend to update the biography reviewing more recent publications. 

Author Response

Rebuttal letter to referee 2:

This referee is highly supportive of publication of our review in IJMS and writes the following: “The manuscript with the title “Insights into active site cysteine residues in Mycobacterium tuberculosis enzymes: Potential targets for anti-tuberculosis intervention” is a study carried out on vital enzymes of the bacterium Mycobacterium tuberculosis (Mtb) and how the mechanism of action of these enzymes may be influenced by the action of drugs in the treatment of tuberculosis.

The importance of the study is very high considering the large number of infected people.

The Cys residue from the catalytic site of four enzymes, its role as a nucleophile and the mechanism of action of those enzymes has been studied. It was also noticed how these can be targeted in therapy by specific drugs. The four Cys-enzymes focused are Alkyl hydroperoxide reductase, Dihydrolipoamide dehydrogenase, Aldehyde dehydrogenase and Methionine aminopeptidase. These enzymes are responsible for the viability and adaptability of Mtb and the infection of host organisms.

The involvement of the Cys residue in substrate recognition and bonding, in the mechanism of enzymatic catalysis is explained, clearly resulting in how it can be interfered with anti-tuberculosis drugs. Interference in the enzymatic catalysis mechanism with different chemical agents with an inhibitory role of the enzyme could be the way to finding new anti-tuberculosis drugs.

I think the study is well done. I have some recommendations.”

Reply: We thank the reviewer for reading our paper and give valuable comments.

In addition, the following points are raised:

  1. In chapter 5.2. Significance of active site cysteine in MetAP function and drug targeting strategies, row 347 is discussed in Figure 4(A). This figure cannot be found in the manuscript.

Reply: Thanks for highlighting this, this was a typo, it should have read “Figure 3(A)”. This has been corrected in the paper.

  1. The bibliography is suitable for the topic discussed, but old enough for a review. Unfortunately, only 12 of the total cited references are from the last 5 years. I strongly recommend to update the biography reviewing more recent publications.

Reply: We expanded the reference list and included a number of newer references.